# COVID-19 related knowledge, attitudes, and practices in Indian Population: An online national cross-sectional survey

**Piyoosh Kumar Singh** [1]* , **Anup Anvikar**[2], **Abhinav Sinha**[2]

**1** Ranchi Field Unit, ICMR-National Institute of Malaria Research, Delhi, India, **2** ICMR-National Institute of Malaria Research, Delhi, India

⊚ These authors contributed equally to this work.
* drpksingh45@gmail.com

**Data Availability Statement:** All relevant data are within the paper and its Supporting information files.

## Abstract

### Introduction

This highly contagious zoonotic corona virus (SARS-CoV-2) spread to most parts of the world (200 countries) and created a public health emergency. Due to its novel nature and indistinctness, different sources of information and suggestions were developed to guide the individuals about its transmission and prevent its infection. Responses to the active intervention efforts have posed some relevant questions on population understanding and attitudes toward COVID-19. The present study is aims to assess the COVID-19 related knowledge, attitude, and practices (KAP) in a heterogeneous Indian population.

### Material and methods

501 respondents across India participated in a questionnaire-based online survey from April 2020 to May 2020. The questionnaire incorporated 56 questions about demographic characteristics and KAP dimensions. The mixed (quantitative and qualitative) methods were employed to evaluate KAP dimensions. Descriptive analysis was estimated as means, SD, and proportion. The bivariate ($\chi^2$), correlation, and regression analysis were utilized for the response analysis. In addition, qualitative analysis, including content and thematic analysis were done for open-ended questions.

### Result

High knowledge and positive attitude were reported in more than half of the study population, with a proportion of 58.6% and 62.1%, respectively. Education shows a significant difference in the knowledge and attitude dimensions. The good practice (50.5% of the total population) reported a significant difference in age and gender categories with the test of independence ($\chi2$). Prevention (56.89%) in knowledge domain and risk (17.56%), information-seeking (45.51%), prevention (51.50%), and treatment-seeking (54.29%) in attitude domains recorded low proportion. KAP variables were found in association in Pearson correlation analysis. In logistic regression analysis, knowledge was the strongest predictor for

**Funding:** The research was volunteered, and no external funding was required for the work.

**Competing interests:** The authors affirm that they have zero competing interests.

the positive attitude, whereas attitude was reported as the best predictor for good practice outcome.

## Conclusion

The study presents a moderate level of covid related knowledge, Attitudes, and Practices in Indian population.

## Introduction

The coronavirus disease (COVID-19) is a severe acute respiratory disease that emerged in a patient with viral pneumonia-like symptoms in Wuhan, Hubei Province of China in December 2019 [1]. The pathogen responsible for the infection is termed as acute respiratory syndrome coronavirus-2 (SARS-CoV-2) which is a new form of coronavirus, initially provisionally labelled as novel coronavirus (nCoV) [2]. This highly contagious, zoonotic virus started infections from a small city and spread rapidly to most parts of the world and created a global health emergency. The World Health Organization (WHO) called for a collaborative effort to tackle the situation and declared it a global pandemic on March 12, 2020 [3]. The regularly updated COVID-19 situation dashboard has reported 40, 49,10,528 confirmed cases and 57, 83,776 deaths globally by this deadly virus as of February 12, 2022 [4].

In India, the first case of COVID-19 got reported in Trissur, Kerala, on 27th January in a 20-year lady with a travel history to China [5]. The rapid movement of people from the global epicenters and between the cities facilitated COVID-19 transmission in India, and infection started spreading to the major cities of India. In response to the increase in cases and to break the transmission chain, active government intervention like international travel suspension, contact tracing, containment, and mitigation strategies were initiated. However, despite the proactive measures, the infection continued to rise in different parts of the country and confirmed cases reached to 4,25,86,544as of February 12, 2022 [6].

To prevent the spread, the application of evidence based NPIs, mainly social distancing, personal & respiratory hygiene with sustained public cooperation in the different communities is warranted [7]. The WHO has also issued specific recommendations for the prevention and control of infection in population and healthcare facilities. It includes maintaining hand cleanliness with hand wash using alcohol-based hand sanitizers, face masks, social distancing, crowd avoidance, self-isolation, and medical attention for a person with mild symptoms (Fever, Cough and headache) [8].

The exponential rise in the number of cases has impended and added severe strain on the healthcare system of all the nations, including the developed and developing countries. Initial attempts also suggested that alone the Medicare effort would not be substantial enough to tackle the situation. The developing or underdeveloped countries have faced the additional challenge of inefficient, unprepared health systems to accommodate active patients in medical facilities. The immediate lockdown step was believed as a vital attempt to control the virus transmission and was followed by different countries. The Indian government also planned a strict nationwide lockdown to minimize social contacts and to reduce the virus community spread. This measure added the additional benefit of reducing the burden of the country's health system and provided the time needed for health system preparedness. The measure included a complete restriction on the movement, non-essential activities, and travel. The constantly updated government guidelines regulated these sudden restrictions and generated

public reactions in panic and confusion that promoted long distance travel of migrant workers to their hometown, which potentially increased the infection risk in other cities.

The effectiveness of any anti-contagion measure is grounded upon the understanding of knowledge, attitudes and practices (KAP) at the macro and micro-level, which will drive the intended participation of individuals in these interventions [9]. So, for implementing these interventions effectively, public education is considered one of the most significant efforts that can vastly help, as has been the case regarding MERS [10] or SARS [11]. Therefore, some vigorous actions were taken by the Ministry of Health and Family Welfare (MOHFW) and state health authorities to educate the general public and provide accurate, and authentic information on COVID-19.

Due to the novel nature and indistinctness of COVID-19, different sources of information and suggestions were developed for guiding the individuals about transmission, and precautions regarding prevention of infection. The whole exercise of providing the information and guidelines becomes increasingly challenging with the heterogeneous nature of population nature and variable education status. The unregulated social media and lack of crisis communication offered vast amount of misinformation and deception, which shaped the clouding understanding of COVID-19, panic, and confusion [12]. Despite the extensive and prolonged lockdowns, a decrease in the doubling time of occurrence of cases, and the number of new cases continued to rise in India [13]. Response to these active intervention efforts posed relevant questions of population understanding and attitudes toward COVID-19. We have some learnings from the earlier SARS outbreak, which highlights the association between knowledge and attitudes with panic and emotion, which might affect the preventive efforts to control the spread of infection [14].

To implement the behavioural change interventions, assessment of population knowledge, attitudes and practices (KAP) is a fundamental step, as it determines the community readiness to accept change. It provides the baseline information and better insight to address the knowledge gap, misconceptions, involved practices for the disease and informs the need for amendment in preventive programs and health awareness plans. There is a profusion of KAP-based studies on different diseases, including infectious diseases and studies on the current COVID-19 pandemic with KAP based study design and variable study populations have been published from different countries as well as from India. Still, most of the studies targeted the healthcare communities [15–19], and paucity of general population-based studies was observed. The present study address the need for Knowledge, Attitude and Practice (KAP) with a mixed method (quantitative and qualitative) study design on heterogeneous Indian population. The results and observations of this study attempted to inform the need for active efforts directed on the societal readiness to comply with pandemic control measures.

## Materials and methods

### Participants and procedure

The study was conducted using a cross-sectional questionnaire survey method among the Indian residents. Due to the highly contagious nature of COVID-19 and to avoid physical interactions, a web-based online survey was planned to record the responses. An online google form was created with brief study details, and the link of the survey was shared to available contacts via email and social media (Facebook and WhatsApp), following the convenient sampling method. The participants were also asked to share the survey link with their contacts. Survey responses for the study were solicited from April 2020 to May 2020. For survey respondents, eligibility was defined with criteria; individuals with Indian residency (living in Indian geographical locations), above the age of 18 or older, and consent for participation. In the

study, Individuals below 18 ages, non-Indian and who does not understand the survey language were excluded to participate in the survey. The formal sample size calculation was performed considering the adult population number of India, and the response rate on the KAP study. The proportion of the population having adequate knowledge regarding COVID-19 was assumed to be 50% as no similar study was conducted in the study population, level of significance 5%, and margin of error 5% were considered for calculation and target sample size for the survey was calculated as 385 using formula ("Sample Size = (Distribution of 50%) / ((Margin of Error%/ Confidence Level Score)$^2$)"). Eventually, considering the incomplete and dropout responses, we established the target sample of 500. The mixed (quantitative and qualitative) methodology was employed for the evaluation of KAP dimensions.

## Questionnaire

Survey questions were constructed with available questionnaire construction information and guidelines from WHO [20] (WHO,2008). The questionnaire incorporated four significant dimensions, demography, knowledge & awareness, attitudes and practices. Apart from primary demographic details, respondents were also asked for healthcare facility distance, and Covid-19 positive status. The knowledge and awareness, attitude and practices dimensions were arranged in sequence and in each part, inquiries on information and communications, knowledge, awareness, transmission, cause and symptoms, preventions and treatment, and care-seeking behaviour domain for COVID-19 were made respectively (Fig 1). The complete questionnaire was limited to 45 questions in total with 20 questions on knowledge, 14 questions on attitudes, 11 questions on practices, and additionally, 11 questions were asked for the demographic's details and health information (S1 Appendix). To facilitate a detailed and better response, questions were developed in categorical (one-optional & multi-optional) and open-ended format. The clarity of the questionnaire was tested in a pilot study among ten students and workers to confirm that the target audience understood the questions. The questions and domains were reviewed for suitability, applicability, relevance, and accuracy by experts comprising social scientist, epidemiologist and medical doctor. The perplexing and challenging

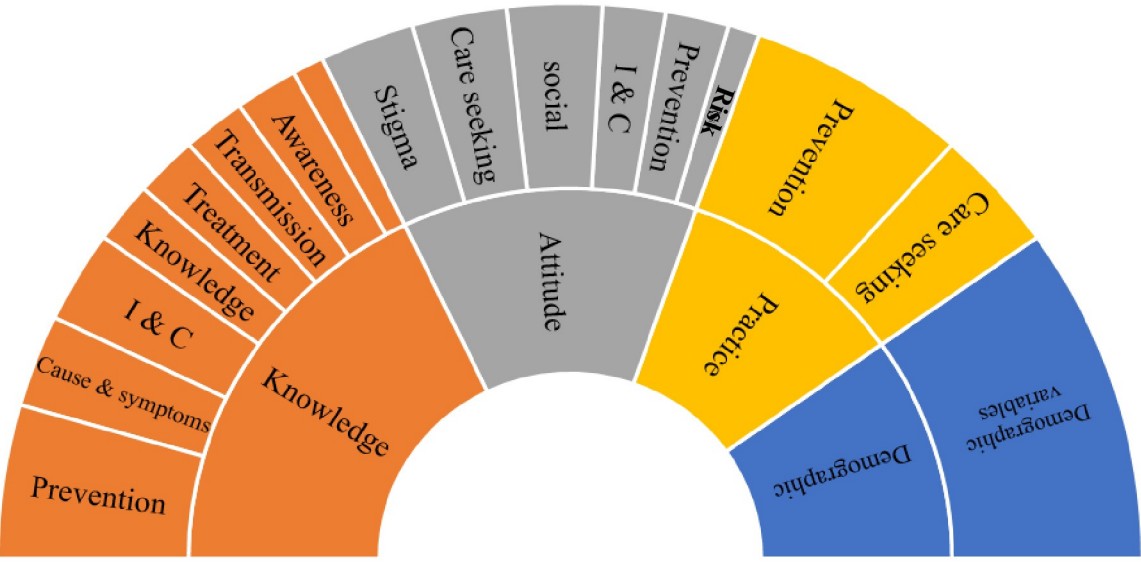

**Fig 1. Questionnaire construct.**

questions were then improved or omitted before the initiation of the study. Data from the pilot study were excluded from the results.

However, pilot feedback assisted to adjust minor alterations and to revise the questions based on the analysis of the comments received. The survey was also tested and modified for length and applicability among the different groups of participants. In order to facilitate completion within an approximate duration of 20 min, the survey was restricted to a total of 56 questions. The translation of the questionnaire was performed by the expert team of Rajbhasha (Hindi) division, National Institute of Malaria research. The Hindi translated version were retranslated to original (English) version, following the backward translation. The two versions were compared for content similarity and no significant differences were observed.

## Scoring and analysis

All the data, including close and open-ended responses, were reviewed and compiled in an window MS Excel sheet for analysis. The scores were calculated for each of the three dimensions of knowledge & awareness, attitude, and practices after assigning respective scores for the responses to the multiple-choice questions in each dimension. One point was awarded for each rightly answered question (for questions with only one correct option) and option (for questions that had more than one correct option) and no point was awarded for wrongly answered questions/options. The knowledge and awareness dimension scores ranged from 0 to 30, for attitudes, the range was 0 to 17, whereas the score range was from 0 to16 for the practice dimension. The total score (63) was calculated after adding all KAP dimensions scores. We followed the published study with a similar study design for the scoring and criteria for assessing the knowledge, attitude, and practices [21]. A cutoff score of $>$ = median value within each dimension was used to categorize the outcome as low, and high (for knowledge and awareness), positive and negative (for attitudes) and good and poor (for practice). Therefore, a participant is said to have: high level of knowledge if the knowledge score was at least 21 points; positive attitude if the attitude dimension was at least 11 points and good practice if the practice score was at least 10 points. Responses to open-ended questions were analyzed with qualitative analysis techniques. Content and thematic analyses were done on the recorded typed/written responses, and appropriate themes or codes were allocated to each response.

## Statistical analysis

The survey response data were captured in Google spreadsheet that was further exported and captured in Microsoft Office Excel 365. Descriptive analysis was done for relevant outcomes and characteristics and summary statistics were presented as means, standard deviations, and frequencies. The chi-square ($\chi^2$) test of independence was performed for comparative analyses of the frequency of participants in different demographic variables. The knowledge, attitude and practice scores and their domain mean scores among the demographic variables were compared with independent samples t-test. The association between KAP categories with different demographic categories were tested using the chi-squared ($\chi^2$) test in bivariate analysis. Logistic regression analysis was planned in bivariate and multivariable models for measuring the magnitude of association between socio-demographic characteristics, and other KAP dimensions. The level of the statistical significance was set for all analyses at $\alpha$ = 0.05. The data visualization charts were developed in Microsoft Office Excel 365. The map was developed with ArcGIS software. All quantitative statistical analysis was done with Special Package for Social Sciences (SPSS) version 23 (IBM). Qualitative analysis, including content and thematic analyses for responses to open-ended questions, were done with NVivo version 12 software.

### Ethical considerations

The protocol for the study was approved by the Institutional Ethics Committee, ICMR-NIMR, New Delhi (PHB/NIMR/EC/2020/106). Participation in the survey was voluntary, anonymous, and in the opening part of the survey, the purpose of the research was communicated in language familiar (Hindi, English) to the respondents. Consent was ensured when a box was checked to designate that the consent was granted. During the survey, participants were provided with the option to deny and withdraw their participation in the survey at any time. Anonymity and confidentiality of participants were ensured during the survey. The participants' details were de-identified, and their responses were stored in a password-protected computer.

## Result

A total of 527 responses were received out of which 13 participants denied consent, ten responses were duplicated, and three were from non-Indians. Following the study protocol, total 501 qualified the study participants' criteria and completed the questionnaire and hence were included in the analysis. The study reported participation from almost all Indian states and union territories. Maximum study participants were from the capital state of Delhi (N = 138, 27.54%), whereas at least one response was received from the newly formed union territory of Ladakh. We could not get any participants from the northeast states of Meghalaya, Tripura, Mizoram, and Sikkim. In terms of geographical distribution, most responses were from north India (301, 60.08%) and least respondents from three northeast states of India (20, 3.99%) (Fig 2).

The socio-demographic characteristics were evaluated with the variables, e.g., gender, age, occupation, education, nearest health facility distance, and residency category of the survey respondents. Data on their demographic details were summarized and presented in Table 1. Of the 501 participants, the majority were male (N = 274, 54.70%), and the mean age recorded was 30.94 ± 10.6 years. Participant's occupation was dominated by the student's category (N = 205,40.90%), followed by individuals involved in government services (N = 156,31.10%). The study group included a large proportion of individuals with a postgraduate university degree or higher education (N = 336,67.10%), and individuals residing in the urban areas (N = 301, 60.10%). The significant portion (N = 449,89.40%) of the respondents reported the 0–10 kilometers home distance from the healthcare facility. To facilitate the statistical analysis the demographic characteristics were further categorized in bivariate categories. The $\chi^2$ goodness of fit test among the categories showed significant differences in all except education characteristics (Table 1).

### Knowledge and awareness assessment

A total of twenty questions with one correct option and more than one correct option were designed to evaluate the knowledge and awareness of COVID-19. The mean knowledge score of the participants was 20.7 with a standard deviation of 3.9. The knowledge and awareness were assessed in domains of cause/symptoms, transmission, prevention, treatment/care-seeking, risk, specific knowledge, and scores were calculated (Table 2). The knowledge & awareness dimension and domain score were also analyzed across different demographic factors. The results demonstrate that knowledge & awareness scores were high in the age group of 31 and above, female gender, working participants, individuals with higher education, and people living in the urban area. Differences in knowledge & awareness dimension among different demographic variables were evaluated using the independent sample t-test. The present study reported a significant difference among education variables in cause/symptoms, transmission, prevention, treatment/care-seeking, specific knowledge and in total knowledge score.

# Participants' Geographical Distribution

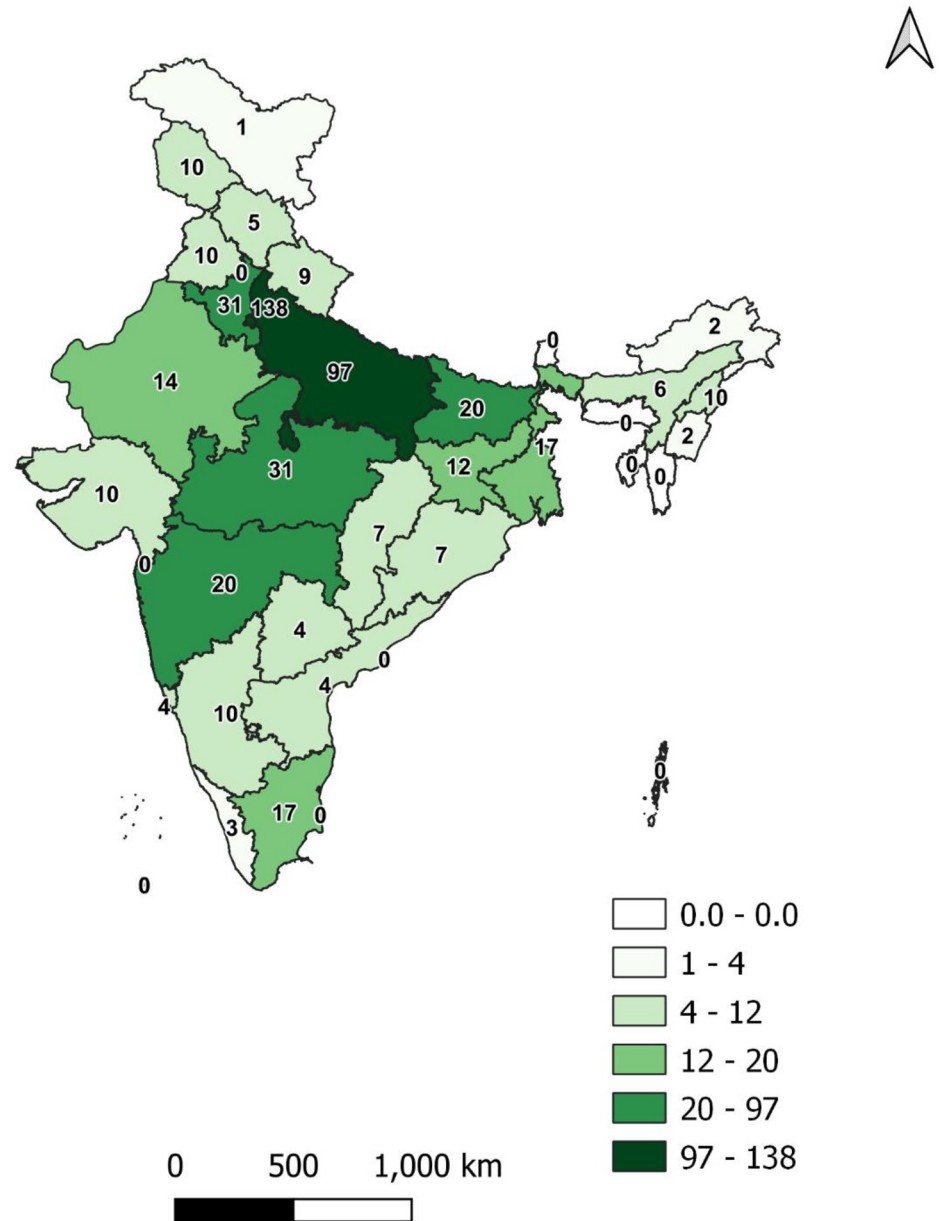

**Fig 2. Participants geographical distribution (number corresoponds to participants from specific locations).**

Similarly, gender and occupation categories show a significant difference in prevention. The t-test results and significant differences were highlighted in different domains as well. The bivariate data and test of significance ($\chi2$) also pointed out the difference in the education factor (Table 2).

**Table 1. Demographic characteristics of the respondents.**

| Characteristics | Category | N | % | Characteristics | Category | N | % |
|---|---|---|---|---|---|---|---|
| **Age** | 30.94 ± 10.6 | | | **Age** | 18–30 | 302 | 60.30% |
| | | | | **(Bivariate)** | Above 30 | 199 | 39.70% |
| | | | | **χ2(p-value)** | 21.176 (0.000) ** | | |
| **Gender** | Female | 227 | 45.30% | **χ2 (p-value)** | 4.409 (.036) * | | |
| | Male | 274 | 54.70% | | | | |
| **Occupation** | Unemployed | 29 | 5.80% | **Occupation** | Not working | 248 | 49.50% |
| | Housewife | 12 | 2.40% | **(Bivariate)** | Working | 253 | 50.50% |
| | Student | 205 | 40.90% | **χ2(p-value)** | 0.050(.823) | | |
| | Workers | 2 | 0.40% | | | | |
| | Business | 6 | 1.20% | | | | |
| | Private job | 73 | 14.60% | | | | |
| | Government Job | 156 | 31.10% | | | | |
| | Healthcare worker | 18 | 3.60% | | | | |
| **Education** | High school (till class 10) | 11 | 2.20% | **Education (Bivariate)** | No higher education | 165 | 32.90% |
| | College (Graduation) | 154 | 30.70% | | higher educated | 336 | 67.10% |
| | Higher education | 336 | 67.10% | **χ2(p-value)** | 58.365(0.000)** | | |
| | (postgraduate or above) | | | | | | |
| **Residence Category** | Rural | 64 | 12.80% | **Residency category** | Non-urban | 200 | 39.90% |
| | Town | 61 | 12.20% | **(Bivariate)** | Urban | 301 | 60.10% |
| | Semi-Urban | 75 | 15.00% | **χ2 (p-value)** | 20.361(0.000)** | | |
| | Urban | 301 | 60.10% | | | | |
| **Health facility distance** | 0–10 kilometers | 449 | 89.40% | **χ2 (p-value)** | 714.71(0.000)** | | |
| | 11–20 kilometers | 32 | 6.40% | | | | |
| | above 20 Km | 20 | 4.00% | | | | |

* χ2 (p-value) significant at the 0.05 level,

** χ2 (p-value) significant at the 0.01 level

## Attitude assessment

Attitude towards COVID-19 was measured by analyzing the responses under the domains of psycho-social, risk, prevention, treatment / care-seeking stigma, and information (Table 3). The attitude score average was 11.2 with SD 2.1,65.88% of the total attitude score (17). The result shows that attitude scores were found to be high in similar demographic variables as were in the knowledge and awareness dimension. In gender, male show a high score as compared to their counterparts. The study also reported a significant difference among the education categories in risk, treatment / care-seeking, and stigma domain score as well as in total attitude score. In the treatment-seeking attitude score. Most of the demographic characteristics showed significant differences in the treatment-seeking attitude score except the residency status. The scores were tested with t-test results and significance scores are marked in the Table 3. The bivariate χ2 test of significance repeated the findings of Knowledge & awareness dimensions and reported the significant difference among the education variables in negative and positive attitude categories (Table 3).

## Practice assessment

The measurement of practice dimensions for COVID-19 was done with inquiries on two major domains: 1) Prevention and 2) Treatment / care-seeking behaviour of participants. The

**Table 2. Knowledge score of the respondents and their demographic distribution.**

| | | Knowledge Domain | Cause / symptoms | Transmission | Prevention | Treatment/ care-seeking | Risk | Knowledge | Total Score | Low (N, %) | High (N, %) |
|---|---|---|---|---|---|---|---|---|---|---|---|
| Age | 18–30 | Mean±SD | 5.33±2.04 | 3.67±1.13 | 7.27±1.4 | 1.85±0.78 | 0.87 ±0.34 | 1.46±0.57 | 20.5 ±4.1 | 78(39.2) | 121 (60.8) |
| | above 30 | Mean±SD | 5.33±2.1 | 3.86±1 | 7.56±1.15 | 1.89±0.67 | 0.92 ±0.27 | 1.47±0.52 | 21±3.5 | 132 (43.7) | 170 (56.3) |
| | | tTest | 0.024 | 1.942 | 2.373 | 0.718 | 1.894 | 0.241 | 0.749 | χ2Test | 1.003 |
| | | p-value | 0.981 | 0.053 | 0.018* | 0.473 | 0.059 | 0.81 | 0.454 | p-value | 0.317 |
| Gender | Female | Mean±SD | 5.38±2.14 | 3.78±1.05 | 7.53±1.19 | 1.79±0.74 | 0.89 ±0.32 | 1.48±0.56 | 20.8 ±3.8 | 92(40.5) | 135 (59.5) |
| | Male | Mean±SD | 5.29±1.99 | 3.72±1.11 | 7.27±1.39 | 1.93±0.73 | 0.89 ±0.31 | 1.46±0.54 | 20.6±4 | 118 (43.1) | 156 (56.9) |
| | | tTest | 0.489 | 0.623 | 2.204 | 2.036 | 0.178 | 0.397 | 1.679 | χ2Test | 0.328 |
| | | p-value | 0.625 | 0.533 | 0.028* | 0.042* | 0.859 | 0.692 | 0.094 | p-value | 0.567 |
| Occupation | Not working | Mean±SD | 5.21±2.02 | 3.69±1.05 | 7.21±1.38 | 1.81±0.78 | 0.86 ±0.35 | 1.46±0.56 | 20.2±4 | 114(46) | 134(54) |
| | Working | Mean±SD | 5.45±2.09 | 3.8±1.11 | 7.57±1.22 | 1.92±0.68 | 0.92 ±0.28 | 1.47±0.54 | 21.1 ±3.7 | 96(37.9) | 157 (62.1) |
| | | tTest | 1.332 | 1.082 | 3.089 | 1.561 | 2.064 | 0.38 | 1.781 | χ2Test | 3.311 |
| | | p-value | 0.183 | 0.28 | 0.002* | 0.119 | 0.040* | 0.704 | 0.075 | p-value | 0.069 |
| Education | No higher education | Mean±SD | 4.94±1.91 | 3.59±1.14 | 7.01±1.57 | 1.73±0.81 | 0.87 ±0.33 | 1.34±0.57 | 19.5 ±4.2 | 81(49.1) | 84(50.9) |
| | higher educated | Mean±SD | 5.52±2.1 | 3.82±1.05 | 7.57±1.12 | 1.93±0.69 | 0.9 ±0.31 | 1.53±0.53 | 21.3 ±3.5 | 129 (38.4) | 207 (61.6) |
| | | tTest | 3.093 | 2.303 | 4.162 | 2.697 | 0.77 | 3.635 | 2.594 | χ2Test | 5.202 |
| | | p-value | 0.002* | 0.022* | 0.000* | 0.007* | 0.441 | 0.000* | 0.010* | p-value | 0.023# |
| Residence category | Urban | Mean±SD | 5.42±2.09 | 3.81±1 | 7.5±1.29 | 1.87±0.73 | 0.91 ±0.29 | 1.45±0.55 | 21±3.7 | 92(46) | 108(54) |
| | Non- Urban | Mean±SD | 5.19±2.01 | 3.66±1.2 | 7.21±1.32 | 1.86±0.74 | 0.86 ±0.35 | 1.49±0.55 | 20.3±4 | 118 (39.2) | 183 (60.8) |
| | | tTest | 1.235 | 1.485 | 2.477 | 0.155 | 1.578 | 0.828 | 0.53 | χ2Test | 2.28 |
| | | p-value | 0.217 | 0.138 | 0.014* | 0.877 | 0.115 | 0.408 | 0.597 | p-value | 0.131 |
| Total | | | 5.33±2.06 | 3.75±1.09 | 7.39±1.31 | 1.87±0.73 | 0.89 ±0.32 | 1.47±0.55 | 20.7 ±3.9 | 210 (41.92) | 291 (58.08) |

* **t-Test** (p Value) significant at the 0.05 level,

** **t-Test** (p Value) significant at the 0.01 level,

# χ2Test (p-value) significant at the 0.05 level

mean score was 10.3 (SD2) for practice outcome, and significant differences were observed among the gender categories with a higher female practice score (10.6 SD 1.9). The bivariate distribution in poor and good practices has reflected the significant prevalence of good practices in 18–30 age cohort and female gender. Other characteristics also showed differences among the categories but could not reach the significance level (Table 4). The response rate for the question, "Would you like to test yourself for Coronavirus infection?" (Yes-36%), and "If there will be a vaccine available against Coronavirus disease (COVID-19), would you consider, having it?" (Yes-75.05%) indicated the moderate attitudes for suggested prevention practices.

The correlation analysis of the Knowledge, Attitude, Practice scores with age reflected a high correlation (Table 5). The data indicated that knowledge score improvement was highly correlated with an increment in a positive attitude, and no correlation was observed with practice. The increase in positive attitude was found to influence the practice in Indian people as

**Table 3. Attitude score of the respondents and their demographic distribution.**

| Demographic Characteristics | | Attitude Domain | Psychosocial | Risk | Prevention | Treatment / care-seeking | Stigma | Information | Total score | Negative (N, %) | Positive (N, %) |
|---|---|---|---|---|---|---|---|---|---|---|---|
| Age | 18–30 | Mean±SD | 2.03±0.7 | 0.17 ±0.38 | 4.47±1.46 | 2.37±0.7 | 1.64 ±0.55 | 0.48±0.5 | 11.2 ±2.1 | 114(37.7) | 188(62.3) |
| | above 30 | Mean±SD | 2.03±0.64 | 0.19 ±0.39 | 4.42±1.58 | 2.54±0.66 | 1.71 ±0.51 | 0.42±0.49 | 11.3±2 | 76(38.2) | 123(61.8) |
| | | t-Test | -0.129 | 0.490 | -0.361 | 2.804 | 1.487 | -1.390 | 0.749 | χ2Test | 0.010 |
| | | p-value | 0.897 | 0.624 | 0.718 | **0.005*** | 0.138 | 0.165 | 0.454 | **p-value** | 0.920 |
| Gender | Female | Mean±SD | 1.95±0.71 | 0.17 ±0.37 | 4.39±1.38 | 2.34±0.7 | 1.7±0.5 | 0.49±0.5 | 11±2 | 88(38.8) | 139(61.2) |
| | Male | Mean±SD | 2.09±0.64 | 0.18 ±0.39 | 4.49±1.61 | 2.52±0.67 | 1.65 ±0.56 | 0.42±0.49 | 11.4 ±2.1 | 102(37.2) | 172(62.8) |
| | | t-Test | -2.373 | -0.441 | -0.753 | -2.918 | 1.147 | 1.566 | -1.679 | χ2Test | 0.125 |
| | | p-value | **0.018*** | 0.660 | 0.452 | **0.004*** | 0.252 | 0.118 | 0.094 | **p-value** | 0.724 |
| Occupation | Not working | Mean±SD | 1.99±0.67 | 0.16 ±0.36 | 4.44±1.43 | 2.33±0.72 | 1.64 ±0.56 | 0.48±0.5 | 11±2.1 | 100(40.3) | 148(59.7) |
| | Working | Mean±SD | 2.07±0.68 | 0.19 ±0.4 | 4.45±1.58 | 2.54±0.65 | 1.7 ±0.51 | 0.43±0.5 | 11.4 ±2.1 | 90(35.6) | 163(64.4) |
| | | t-Test | -1.246 | -1.071 | -0.111 | -3.329 | -1.225 | 1.280 | -1.781 | χ2Test | 1.200 |
| | | p-value | 0.213 | 0.285 | 0.911 | **0.001*** | 0.221 | 0.201 | 0.075 | **p-value** | 0.273 |
| Education | No higher education | Mean±SD | 2.01±2.01 | 0.22 ±0.22 | 4.35±4.35 | 2.24±2.24 | 1.56 ±1.56 | 0.49±0.49 | 10.9 ±10.9 | 74(44.8) | 91(55.2) |
| | Higher educated | Mean±SD | 2.04±0.65 | 0.15 ±0.36 | 4.5±1.49 | 2.53±0.62 | 1.72 ±0.51 | 0.44±0.5 | 11.4±2 | 116(34.5) | 220(65.5) |
| | | t-Test | -0.554 | 2.007 | -1.057 | -4.173 | -3.061 | 1.127 | -2.594 | χ2Test | 5.011 |
| | | p-value | 0.580 | **0.045*** | 0.291 | **0.000*** | **0.002*** | 0.260 | **0.010*** | **p-value** | **0.025**# |
| Residency category | Non-urban | Mean±SD | 2.1±0.65 | 0.18 ±0.39 | 4.34±1.46 | 2.45±0.68 | 1.65 ±0.57 | 0.45±0.5 | 11.2 ±2.2 | 74(37) | 126(63) |
| | Urban | Mean±SD | 1.99±0.69 | 0.17 ±0.38 | 4.52±1.54 | 2.43±0.7 | 1.69 ±0.51 | 0.46±0.5 | 11.3±2 | 116(38.5) | 185(61.5) |
| | | t-Test | 1.759 | 0.208 | -1.296 | 0.341 | -0.876 | -0.369 | -0.530 | χ2Test | 0.121 |
| | | p-value | 0.079 | 0.835 | 0.195 | 0.733 | 0.381 | 0.712 | 0.597 | **p-value** | 0.728 |
| Total | | | **2.03±0.68** | **0.18 ±0.38** | **4.45±1.51** | **2.44±0.69** | **1.67 ±0.53** | **0.46±0.5** | **11.2 ±2.1** | **190(37.9)** | **311(62.1)** |

* **t-Test** (p Value) significant at the 0.05 level,

** **t-Test** (p Value) significant at the 0.01 level,

# χ2Test (p-value) significant at the 0.05 level

data reflects. The analysis of the KAP dimensions score with age indicates that the knowledge about COVID-19 had increased with increasing age (Table 5).

In the present study, the bivariate and multivariate logistic regression analysis were applied to identify the predictors for a positive attitude and good health practice outcomes. The analysis, used demographic variables (age, gender, occupation, education, residence), knowledge, attitudes, and practice scores as independent variables. The result of the analysis, presented in Table 6, highlights a significant association of knowledge and practice score with positive attitude outcome, whereas age (above 30), gender (male), and attitude variables presented significant contribution in the development of good health practice. The association was checked in both bivariate and multivariate regression models and similar findings were observed. Odds ratio with confidence interval values in unadjusted and adjusted patterns indicate the results from bivariate and multivariate models, respectively (Table 6).

**Table 4. Practice score of the respondents and their demographic distribution.**

| Demographic Characteristics | | Practice Domain | Prevention | Treatment / care-seeking | Total score | Poor (N, %) | Good (N, %) |
|---|---|---|---|---|---|---|---|
| Age | 18–30 | Mean±SD | 9.29±1.76 | 1.13±0.72 | 10.4±2 | 134(44.4) | 168(55.6) |
| | above 30 | Mean±SD | 9.07±1.64 | 1.05±0.72 | 10.1±1.9 | 114(57.3) | 85(42.7) |
| | | t Test | -1.414 | -1.154 | -1.652 | χ2Test | 8.005 |
| | | p-value | 0.158 | 0.249 | 0.099 | p-value | **0.005**\* |
| Gender | Female | Mean±SD | 9.48±1.54 | 1.14±0.7 | 10.6±1.9 | 95(41.9) | 132(58.1) |
| | Male | Mean±SD | 8.97±1.81 | 1.06±0.73 | 10±2 | 153(55.8) | 121(44.2) |
| | | t Test | 3.425 | 1.283 | 3.408 | χ2Test | 9.719 |
| | | p-value | **0.001**\* | 0.2 | **0.001**\* | p-value | **0.002**# |
| Occupation | Not working | Mean±SD | 9.27±1.78 | 1.1±0.73 | 10.4±2 | 115(46.4) | 133(53.6) |
| | Working | Mean±SD | 9.13±1.65 | 1.09±0.71 | 10.2±2 | 133(52.6) | 120(47.4) |
| | | t Test | 0.913 | 0.154 | 0.85 | χ2Test | 1.925 |
| | | p-value | 0.362 | 0.877 | 0.396 | p-value | 0.165 |
| Education | No higher education | Mean±SD | 9.28±1.71 | 1.08±0.74 | 10.4±2 | 79(47.9) | 86(52.1) |
| | higher education | Mean±SD | 9.16±1.72 | 1.1±0.71 | 10.3±1.9 | 169(50.3) | 167(49.7) |
| | | t Test | 0.743 | -0.239 | 0.559 | χ2Test | 0.259 |
| | | p-value | 0.458 | 0.811 | 0.576 | p-value | 0.611 |
| Residency category | Non-urban | Mean±SD | 9.11±1.81 | 1.09±0.72 | 10.2±2 | 99(49.5) | 101(50.5) |
| | Urban | Mean±SD | 9.27±1.65 | 1.1±0.72 | 10.4±1.9 | 149(49.5) | 152(50.5) |
| | | t Test | 0.743 | -0.239 | 0.559 | χ2Test | 0 |
| | | p-value | 0.458 | 0.811 | 0.576 | p-value | 1 |
| Total | | | **9.2±1.71** | **1.1±0.72** | **10.3±2** | **248(49.5)** | **253(50.5)** |

\* **t-Test** (p Value) significant at the 0.05 level,

\*\* **t-Test** (p Value) significant at the 0.01 level,

# χ2Test (p-value) significant at the 0.05 level

## Information and communication

The study assessed the Information and communication domain with four questions (3-Knowledge and awareness, 1- Attitude) responses: **1)** When did you first hear about coronavirus? **2)** Where/from whom did you first hear about coronavirus? **3)** Whom do you trust to give you accurate information about Coronavirus disease (COVID-19)? **4)** If you need to know about coronavirus disease, what would you like more information about?

**Table 5. Correlation matrix among knowledge, attitude and practice score and age.**

| | | Age | Knowledge Score | Attitude score |
|---|---|---|---|---|
| **Age** | **Pearson Correlation** | | | |
| | **p-value (2-tailed)** | | | |
| **Knowledge Score** | **Pearson Correlation** | **.095**\* | | |
| | **p-value (2-tailed)** | **.034** | | |
| **Attitude score** | **Pearson Correlation** | .056 | **.327**\*\* | |
| | **p-value (2-tailed)** | .212 | **.000** | |
| **Practice score** | **Pearson Correlation** | -.029 | .098\* | **.176**\*\* |
| | **p-value (2-tailed)** | .517 | **.028** | **.000** |

\*.Pearson Correlation significant at the 0.05 level (2-tailed)

\*\*. Pearson Correlation significant at the 0.01 level (2-tailed)

**Table 6. Regression analysis among KAP variables and demographic characteristics.**

| Variables | Unadjusted | | | Adjusted | | |
|---|---|---|---|---|---|---|
| | Wald | Odds Ratio (Upper-Lower) | p-value | Wald | Odds Ratio (Upper-Lower) | p-value |
| **Attitude** | | | | | | |
| **Age** | 0.01 | 0.981(0.679–1.419) | 0.92 | 0.597 | 0.824(0.504–1.347) | 0.44 |
| **Gender** | 0.125 | 1.067(0.743–1.533) | 0.724 | 0.596 | 0.854(0.784–1.748) | 0.44 |
| **Occupation** | 1.199 | 0.8172(0.569–1.173) | 0.273 | 0.424 | 0.85(0.522–1.386) | 0.515 |
| **Education** | 4.983 | 0.648(0.443–0.9484) | 0.0256 | 0.912 | 0.816(0.538–1.238) | 0.34 |
| **Residency** | 0.121 | 1.068(0.738–1.544) | 0.728 | 0.82114923 | 0.365(0.804–1.81) | 0.365 |
| **Knowledge score** | 38.300 | 1.183(1.122–1.247) | **0.000**** | 33.742 | 1.181(1.117–1.25) | **0.000**** |
| **Practice Score** | 9.837 | 1.16(1.057–1.273) | **0.002**** | 8.249 | 1.161(1.049–1.286) | **0.004**** |
| Practice | | | | | | |
| **Age** | 7.954 | 1.681(1.172–2.413) | **0.005**** | 5.4 | 0.582(0.368–0.919) | **0.020*** |
| **Gender** | 9.653 | 1.757(1.231–2.507) | **0.002**** | 9.412 | 1.805(1.238–2.633) | **0.002**** |
| **Occupation** | 1.922 | 0.78(0.549–1.108) | 0.166 | 0.341 | 0.874(0.555–1.375) | 0.559 |
| **Education** | 0.259 | 0.908(0.625–1.317) | 0.611 | 0.286 | 1.116(0.746–1.671) | 0.592 |
| **Residency** | 0 | 1(0.699–1.43) | 1 | 0.045 | 1.042(0.713–1.522) | 0.831 |
| **Knowledge score** | 1.687 | 1.031(0.985–1.079) | 0.194 | 0.062 | 1.007(0.956–1.06) | 0.803 |
| **Attitude Score** | 11.095 | 1.16(1.063–1.264) | **0.001**** | 11.743 | 1.178(1.073–1.294) | **0.001**** |

* Significant at the 0.05 level

** significant at the 0.01 level

All the respondents mentioned familiarity with the term "coronavirus," and most of the respondents had listened the word in recent months (65%). The most common sources of their information on COVID-19 were the print media (26.31%) and internet/social media (23.79%). The share of other communication mediums was also evaluated (Fig 3). The trusted sources of information for the Indian population were the National (25.3), International health agencies (21.41%), followed by the healthcare workers (14.28%). Th addition, respondents expressed their wish to know more about COVID-19, mostly on the treatment options (20%), prevention (17%), Government actions for prevention (15%) and signs & symptoms (15%) (Fig 3(d)).

## Health behaviour and barriers

Health-related behaviour and barriers were also assessed with the survey response dataset and presented in Fig 4. Participants conveyed the challenges in necessary item access (33%) and financial inability (12%) to procure the resources in taking preventing actions for COVID-19. Assessment of treatment-seeking behaviour in the last 30 days demonstrated that half of (54.33%) fever patients (more than three days) believe in going to the hospitals. However, in general, the Indian population, hospital visiting rate for the treatment was 21.16% (Fig 4).

## Qualitative analysis

Content analysis with word frequency query for most used words was performed to ascertain the distinct knowledge about the reason for lockdown measures. Cloud image with 20 most frequent words was developed and reported the popular term used by the participants to define the reason, e.g., Spread, Virus, Corona, Prevent, Covid, distancing, transmission etc. Further, thematic based qualitative analysis was employed to analyze the open-ended question

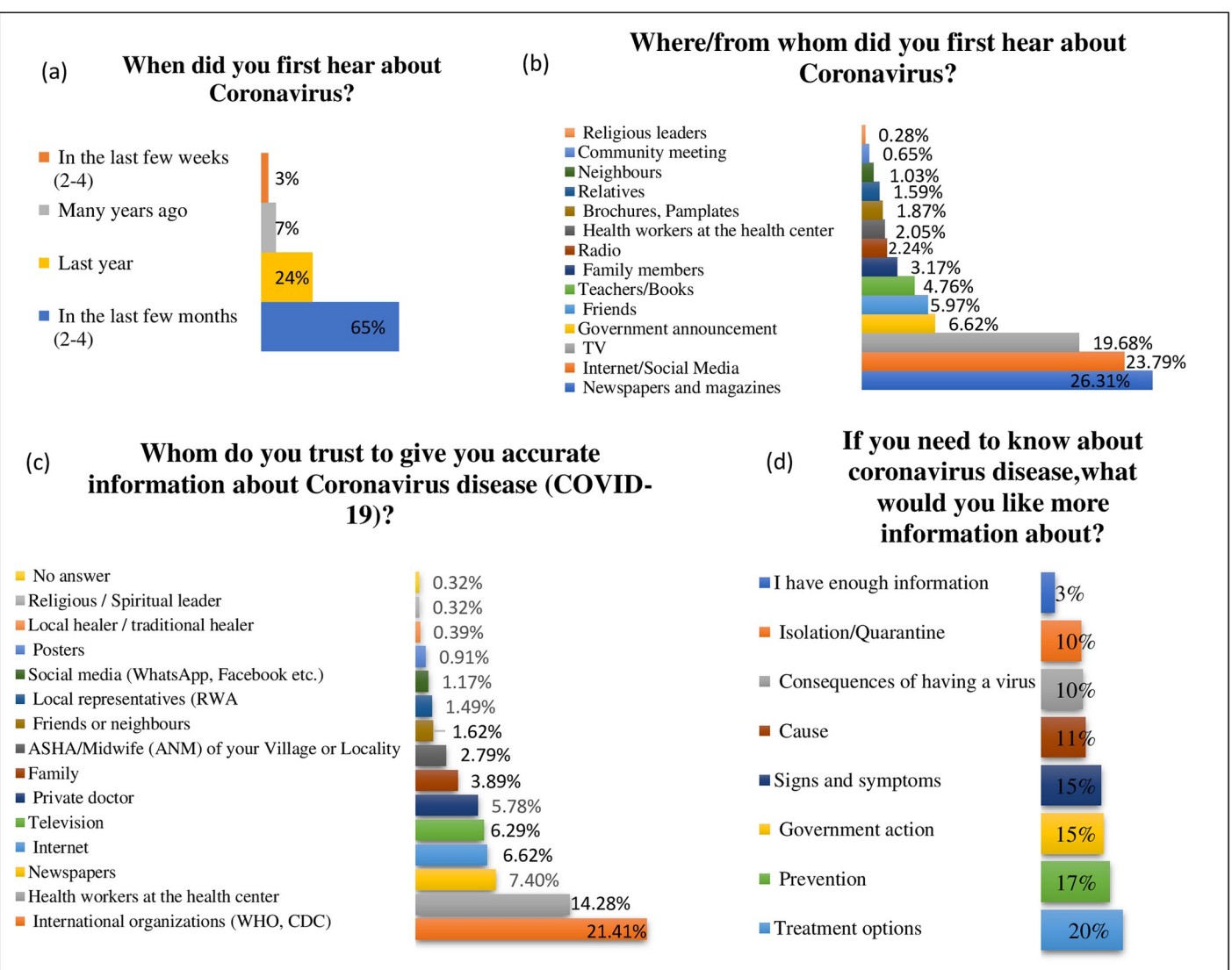

**Fig 3. Information and communication.** (a) First Information time (b) Mode of information & communication (c) Reliable sources for information & communication (d) Information & communication need.

response for the inquiry on "most effective ways of preventing Coronavirus disease (COVID-19) or infection". Each response was coded with the listed themes (Fig 5b) and a frequency distribution table was created to find out the most favored way. The large section of the study population supported the mode of social distancing, followed by the washing hand with soaps. The respondents also realized the need for authentic information; however, not at a high rate.

The assessment of transmission source perception for COVID-19 was explored with the response of an open-ended question; "In your opinion, who is responsible for the transmission of Coronavirus disease (COVID-19) in your community/region? The responses were analyzed, and some major themes were identified. Most of the respondents (32.69%) wrote about the people with irresponsible behaviour (32.96%), e.g., Do not follow the lockdown rules, unnecessary going outside. Some respondents (9.22%) also discussed the role of authorities considering lockdown measures. The need for improved knowledge and awareness was also felt by some (8.38%) respondents, as highlighted in the "lack of proper information" theme.

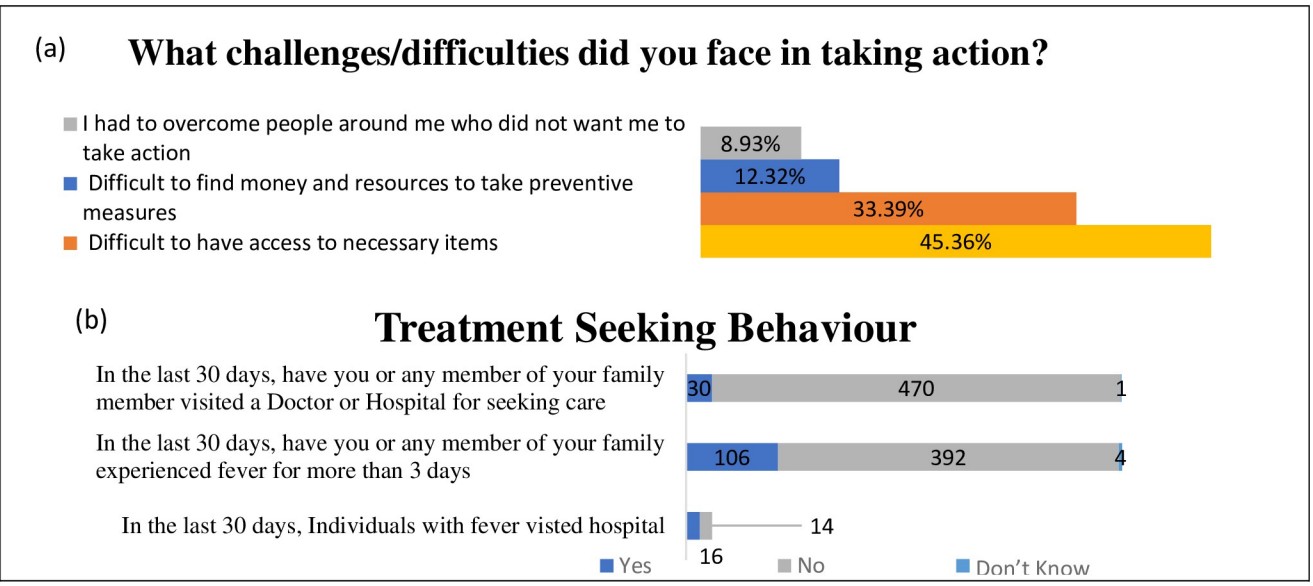

**Fig 4. Practice behavior. (a)** Barrier for preventive practices (**b**) Treatment/care seeking behaviour during COVID-19 pandemic.

## Discussion

The present survey evaluated community knowledge, attitude, practices towards COVID-19 among a heterogeneous, adult and the literate population of India that could be reached during the lockdown via the internet only. The survey was conducted during the initial phase of the pandemic impact in India. Realizing the emergent need, some recent studies have been published on knowledge, attitudes and practices towards COVID-19 from India [22, 23] and other parts of the countries, majorly among the health care workers [15–19] and also in the general population [24–28]. The novelty of the study is to ascertain the public knowledge, attitudes, and practices towards COVID-19 with a wide range of domains in a heterogeneous Indian population. The respondent population ensured participation from most Indian states and a preponderance of (301, 60.08%) North Indian population was observed. However, the limited share (36%) of internet users in the Indian population has restricted our scope of involving more wide-ranging participants [29]. So only educated people and those who understand English or Hindi and had internet access were included in the study.

The mean total KAP score of 42.2 SD 5.6 (out of 63) for the study population was recorded, and the median (43) based categorization indicated half of the population (51.8%) with good knowledge, attitude, and practice behaviour. The demography of the respondents shows that a large portion of adults (age 18–30), males (54.70%), individuals involved in some kind of jobs (50.50%) (though student proportion (40.90%) was noteworthy), educated with a higher degree (67.10%) and individuals living in urban (60.10%) parts of India responded the survey more. The skewness of the study population was presumed as only literate, and internet user population could access the survey due to the inherent limitations as mentioned above. The difference among the demographic variables was tested and found significant in most characteristics except in occupation. This study revealed that nearly all the (98.60%) respondents know about the causes of the COVID-19 crisis. All the respondents agreed to be familiar with the word "coronavirus" but denied any infections in their family. In the present study, the mean knowledge score for the study population was **20.7 SD 3.9** (Total Knowledge score = 30). The knowledge dimension assessed he knowledge about the characteristics of the

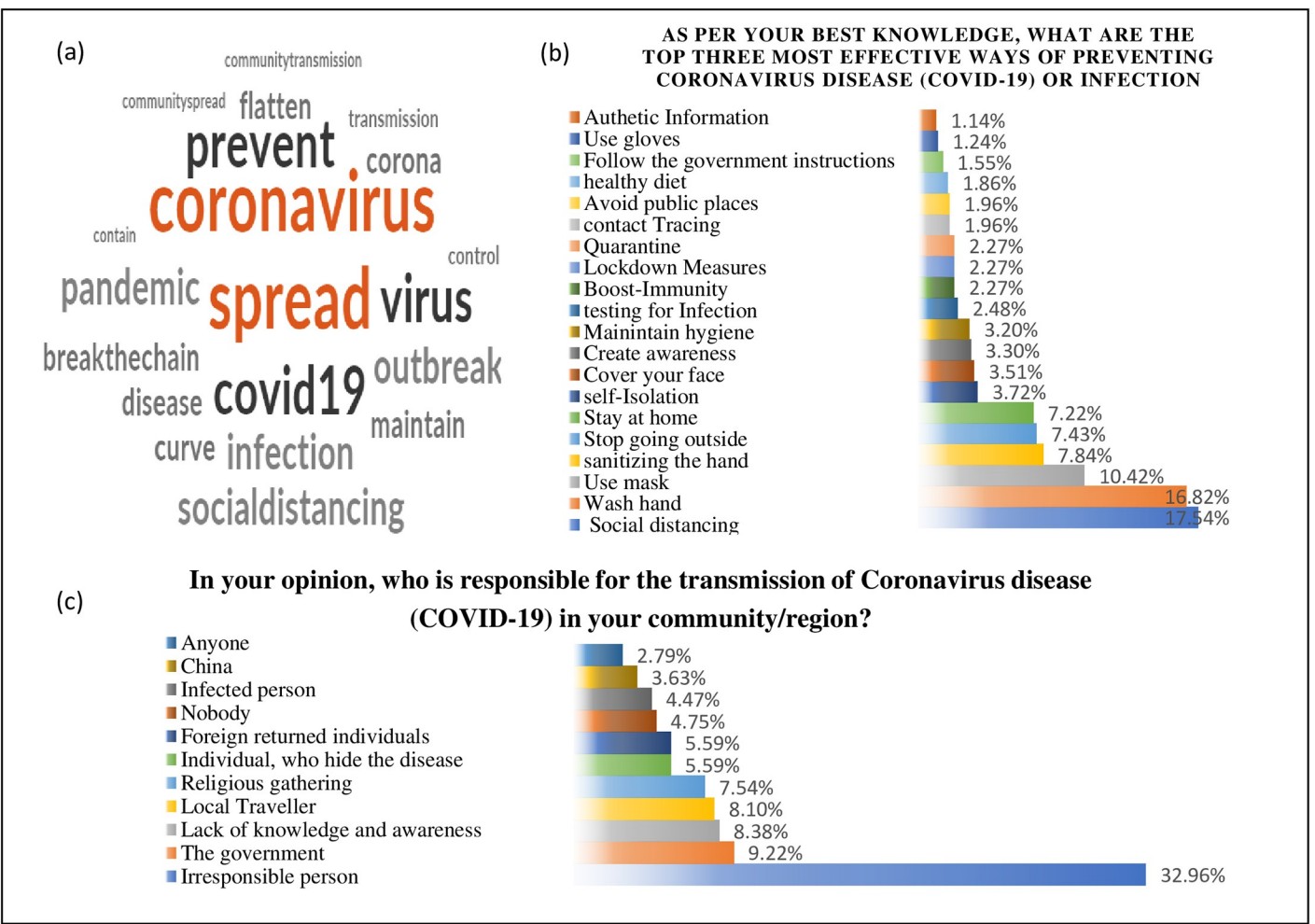

**Fig 5. Qualitative analysis. (a)**- Word cloud of terms (words) used to describe the reason of lockdown **(b)**—Identified themes and their frequency for reported preventive measures **(c)**—Identified themes and their frequency for 'transmission source'.

COVID-19, the mode of transmission and prevention against virus transmission primarily. Approximately 58% of respondents were found in the high knowledge category whereas the domains; specific knowledge (97%), and risks (89%) recorded high response in comparison to the domain's cause/symptoms (63%), transmission (75%), prevention (57%), and treatment /care-seeking (70%). Following the results, the present study warrants the need for prevention-, and treatment-specific knowledge communication among the Indian population. The degree of knowledge about COVID-19 reported in this study is higher to the Bangladeshis (33%) [30], comparable to the KAP studies on the Indian general population (50%) [23], Iraq (52.15%) [31], Syria (60%) [32], health care workers in Uganda [17] and lower than that reported in China (90%) [28], Indian population (83.7%, 81%) [33, 34], Tanzania (77%) [27] and the US (80%) [35] general population. Similarly, lower than the healthcare population of India (71.2%) [22] Vietnam Healthcare Workers (88.4%.) [36] and in Multinational healthcare worker (92.7%) [37]. The knowledge score and high knowledge proportion was significantly higher in higher educated individuals, and the knowledge about prevention was found to be significantly different in all measured demographic characteristics. The bivariate analysis in

the knowledge dimension pointed out the need for more diverse awareness initiative to educate differently educated people.

The study estimated that about 6 in 10 (62.1%) Indians have a positive attitude about COVID-19. The finding is comparatively higher than the Iraq population (22.2) [31], Indian construction worker population (32.8) [38] and lower than the Vietnam Healthcare population (90%) [36], Indian (77.33, 77%) [33, 34], Chinese [28], and Malaysian general population [24]. The attitude score of the study group was **11.2 SD 2.1** (65.88% of the total attitude score), which was significantly more in higher educated people. The domain categories recorded a small proportion of positive attitudes in risk perception (18%), information seeking attitude (46%), prevention (51%), and treatment-seeking attitude (54%). The study population scored better for psychosocial attitude (82%) and stigma towards COVID-19 patients (70%). The difference in treatment-seeking attitude was found significant in different demographic variables. The perception of risk assessed with the question "you or your family members will get Coronavirus disease (COVID-19) in the next 1–2 months" and moderate attitude for the current healthcare system had given some evident suspicion about the high transmission of SARS-CoV-2 in India. A 22 countries-based study reports the fair knowledge in many countries and good attitudes among the residents of many countries, but significant portion of the studied population reflected little knowledge about the symptoms and treatment [39]. The knowledge and attitude dimensions finding suggest developing a more curated awareness and communication program for COVID-19 with differential targeting.

In agreement with the previous study [18], our study reported a significantly high COVID-19 related practice score among the females. In other screened demographic variables, younger age groups, lesser educated people, non-working respondents, and people living in urban areas showed good practice scores but were not statistically significant. Study findings affirm that only half of the studied (50.5%) population is involved in good practice behaviour towards COVID-19, which is comparable to Iraq general population (46.2%) [31] and lower than the reported in multinational healthcare workers (79%) [32] and Indian general population (63.8) (83.5%) [33, 34]. A study on Indian construction workers presented lower (20.2) practice (social distancing) to our practice findings [38]. The treatment-seeking behaviour of the participants with "fever (more than three days) in the last 30 days" indicates the same proportion (53%) (Fig 4(b)). Participants in the study showed low (36%) compliance for testing and a high (74%) positive response for future vaccines use. The correlation matrix confirms a significant positive correlation among knowledge-attitude, and attitude-practices, which coincides with the previous studies findings and suggest that interventions should target the knowledge promotion and increasing positive attitude to improve the covid-19 related preventive practices [39]. The increasing age was associated with the knowledge score. The logistic regression analysis model allowed us to quantify the effects of demographic and KAP variables in developing positive attitudes and good practices. The present study highlights the role of age (30 and above), gender (Male), and attitude (Positive) as active predictors for the development of COVID-19 related practice outcomes as tested in the bivariate and multivariate model. Positive attitudes have shown a strong association (p = 0.000) with knowledge compared to practice (p = 0.004). The study suggests that most of the participants followed the credible mode and source of information; however, the role of another active medium like social media is also evident and needs to be evaluated in terms of communicating misinformation, unverifiable content, and creating misconception [40] The Study on Ebola pointed out the role of, the trust deficit for the healthcare system as well as belief in misinformation which had created a decrease in adopting preventive behaviour [41]. This misinformation can also be attributed to the non-acquaintance and lack of training in crisis communication methodology and principles. Customized IEC campaigns to the target populations with the local terminology can

reduce the chances of misinformation and misconception among the different communities. Only half of the population (53%) with three days of fever have sought hospital or doctor-based care in the last 30 days. Qualitative analysis of the open-ended question identifies the key themes and endorses increasing awareness about the suggested prime preventive measures (social distancing, handwash, and mask use) among the participants. The study outcome specifies the active adaptation of high knowledge into a positive attitude (71%) but a comparatively lower conversion of a positive attitude to good practice (56%). Only half of the individuals (50%) with high knowledge showed good practice for COVID-19, suggesting the lower progression of knowledge-to-practice compared to attitude-to-practice.

The study results demonstrate that almost a quarter (119, 24%) of Indian literate people had Covid-19 related high knowledge, positive attitude and good practice. The information arriving in any community may not necessarily become a part of their knowledge and wisdom, and variations in knowledge and perception are apparent. This synchronization needs to be understood with their believes, traditions and practices of the community. The public has access to many sources of information, and they form their perceptions based on these diverse sources and not merely on official sources. These perceptions regulate the attitude to add significant stimulus to perform various practices like social distancing, isolation, quarantine, mask use, personal hygiene and empathy to patients and healthcare workers. The volatile existence of deadly viruses with ill-defined transmission behavior and the lack of awareness often influences the readiness to address these challenges unexpectedly. The study presents some additional advantages over comparable studies from different parts of the world due to the inclusion of multi-optional and open-ended questions for better expression of multilingual, and multicultural respondents. The present KAP inquiry attempted to address the Indian population variation in terms of education status, socio-economic status, and high heterogeneity, and the study inferences should not be viewed as a representative reflection of all Indian communities. The central suggestion of the study lies with the fact that participation at the individual level should be ensured for COVID-19 control with active efforts to increase the knowledge, attitude and practices on COVID-19. These efforts will be operative for achieving this great goal of humanity free of the SARS-CoV-2 (coronavirus) and prevent us from future such infections.

## Limitations

There are some potential limitations in the study that need to be highlighted. The snowball and convenient sampling procedure could have introduced systemic unavoidable selection bias. The scoring pattern of the questions was the same for all the domains of different KAP dimensions. The response data in this study are self-reported and depend on the participants' honesty & recall ability and may create recall bias. The questionnaire's pattern and mix (quantitative and qualitative) design did not allow us for validity and reliability statistical analysis. Most respondents for the survey were from literate and urban, semi-urban locations, which may not truly represent the varied Indian population and participation from marginalized, neglected, and underprivileged populations could have been ensured.

## Conclusions

In summary, the present study demonstrates moderate level of knowledge (High), positive attitudes and good practices about COVID-19 among Indians. The mode of information and its access to the population are good but domain specific education initiatives may be adopted to improve knowledge and preventive practices for COVID-19 infections. In future, more comprehensive (e.g., qualitative) studies should be designed in a better inclusive sample from the

different heterogeneous groups. Even with some limitations, our findings present valuable information about the KAP dimensions and their dynamics in the Indian population.

## Supporting information

**S1 Appendix. Knowledge, attitude, and practice questionnaire on coronavirus disease (Covid-19).**
(DOCX)

**S1 File. KAP data.**
(CSV)

## Acknowledgments

We are thankful to all the respondents for their voluntary participation providing their contribution into the study. Without the assistance of our colleagues Dr. Vandana Sharma, Rajbhasha Division (ICMR-NIMR Delhi), the execution of the translation work could not have been feasible. We would also like to express our appreciation to the colleagues from ICMR-NIMR and friends for their kind assistance in data collection.

## Author Contributions

**Conceptualization:** Piyoosh Kumar Singh, Abhinav Sinha.

**Data curation:** Piyoosh Kumar Singh, Abhinav Sinha.

**Formal analysis:** Piyoosh Kumar Singh.

**Investigation:** Piyoosh Kumar Singh, Abhinav Sinha.

**Methodology:** Piyoosh Kumar Singh, Anup Anvikar, Abhinav Sinha.

**Project administration:** Piyoosh Kumar Singh.

**Software:** Piyoosh Kumar Singh.

**Validation:** Piyoosh Kumar Singh, Anup Anvikar, Abhinav Sinha.

**Visualization:** Abhinav Sinha.

**Writing – original draft:** Piyoosh Kumar Singh.

**Writing – review & editing:** Piyoosh Kumar Singh, Anup Anvikar, Abhinav Sinha.

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
