## [Decision Letter · Decision Letter 0]

26 Dec 2021

PONE-D-21-12498COVID-19 related knowledge, attitudes, and practices in Indian Population: An online national cross-sectional surveyPLOS ONE

Dear Piyoosh Kumar Singh

Thank you for submitting your manuscript to PLOS ONE. After careful consideration, we feel that it has merit but does not fully meet PLOS ONE’s publication criteria as it currently stands. Therefore, we invite you to submit a revised version of the manuscript that addresses the points raised during the review process.

We look forward to receiving your revised manuscript.

Kind regards,

Muhammad Junaid Farrukh

Academic Editor

PLOS ONE

Additional Editor Comments (if provided):

Dear Author,

after carefully reviewing your manuscript and reviewers comments, you are required to do major revision in your manuscript and submit again.

Journal Requirements:

3. Please include additional information regarding the survey or questionnaire used in the study specifically please include a copy, in  the original language

5. Please ensure that you refer to Figure 3 in your text as, if accepted, production will need this reference to link the reader to the figure.

6. We note that Figure 2 in your submission contain map images which may be copyrighted. All PLOS content is published under the Creative Commons Attribution License (CC BY 4.0), which means that the manuscript, images, and Supporting Information files will be freely available online, and any third party is permitted to access, download, copy, distribute, and use these materials in any way, even commercially, with proper attribution. For these reasons, we cannot publish previously copyrighted maps or satellite images created using proprietary data, such as Google software (Google Maps, Street View, and Earth). For more information, see our copyright guidelines: http://journals.plos.org/plosone/s/licenses-and-copyright.

7. We note you have included a table to which you do not refer in the text of your manuscript. Please ensure that you refer to Table 4 in your text; if accepted, production will need this reference to link the reader to the Table.

Reviewers' comments:

Reviewer's Responses to Questions

**Comments to the Author**

1. Is the manuscript technically sound, and do the data support the conclusions?

Reviewer #1: Yes

Reviewer #2: Yes

Reviewer #3: No

2. Has the statistical analysis been performed appropriately and rigorously? 

Reviewer #1: No

Reviewer #2: Yes

Reviewer #3: No

3. Have the authors made all data underlying the findings in their manuscript fully available?

Reviewer #1: Yes

Reviewer #2: Yes

Reviewer #3: Yes

4. Is the manuscript presented in an intelligible fashion and written in standard English?

Reviewer #1: Yes

Reviewer #2: Yes

Reviewer #3: No

5. Review Comments to the Author

Reviewer #1: I did not see any reliability and validity test. We should know the Cronbach alpha value for this questionnaire. Would you please Provide the flow chart of your research? I did not see the objective in the abstract, And in methodology, please mention the exclusion criteria. In the result, please mention in of table below the statistic that you used. I saw some paragraphs did not mention the table

For discussion, please add more the similar study in others country.

In conclusion, please make sure the statement based on the objective of the study

Reviewer #2: Overall found this to be a good read and well done research taking into account the circumstances of the pandemic at the time of the study. That shows forward-thinking and adaptability.

Introduction

second paragraph. Kindly add the year 2020 to the 27th January to prevent any ambiguity regarding the time although it is apparently obvious. It does not hurt to add it.

Last sentence of paragraph 2; would be good to have the figures as of the time of this article being published or a recent update in 2021. Also "....as on August..." should be "...as of August.."

third paragraph.

Kindly rephrase this to reflect that you refer to that period. Maybe you could change "so far" to "At that time" and "have been developed" to "had been developed" as well as the subsequent tense of the sentence to match.

Regarding the situation during the study period, kindly change the tense to the past tense like above to reflect what was happening at that time. In this current state it sounds like that is the current situation in India.

METHODS

Line 6: participants were also asked to share the survey link with "their" instead of "his"

Questionnaire

line 5: you mention infection status. I think this would be better if defined as it leaves room for subjectivity. What does this mean for a participant who selects yes or no. Is their answer based on official results from the country's testing program or from symptoms the patient has or some other means.

Scoring and analysis

State the basis of the scoring system used. Was it arbitrarily generated or based on for example WHO's questions you developed your questionnaires on? It is important that this is very clear as your entire study seems to be based on this scoring system.

RESULTS

the scores are mentioned again in a manner i believe not to be intuitive. Clarifying the scoring system will help this. Also state what the overall score would have been again to make it easy for the reader to follow. Kindly review this in the results section.

KNowledge and Awareness assessment

Line 7: the results demonstrate that knowledge scores.... The first part can be removed to make the sentence less wordy and read better. It could be "knowledge scores were found to

be high in the age group of 31 and above, female gender, working participants, individuals with

higher education, and people living in the urban area."

Discussion

The first paragraph provides context and the need for your work. This belongs in the introduction as justification. The discussion should begin with key findings of the study

Limitations should also come before the conclusions

Reviewer #3: Dear Authors,

Please kindly refer to my specific comments for improvement of the manuscript below:

1. Keywords in abstracts - There are a few keywords that I find unnecessary e.g. awareness and survey. Please kindly revise

2. Spelling - please standardize the spelling for 'coronavirus' vs 'corona virus'.

3. Methodology - Please refer to the questions below:

- Was there any target population planned before the distribution of the survey? I understand that the survey was distributed in Facebook and Whatsapp. However, there must be target groups that you were trying to reach. Please clarify.

- Has the questionnaire undergone reliability and validity tests? If yes, please include the Cronbach's alpha values.

- Who performed the questionnaire translation? Did the translation process follow a standard protocol? Please elaborate.

- Please revise the total number of questions to 45 and not 56 as the other 11 questions were related to demographics.

- What does the term 'dropout' response refer to here? I personally think that the term is not applicable in a survey-type research.

- What was the formula used to calculate the sample size in this research? Please clarify and think of the need to calculate the sample size for this type of research.

4. Results - the overall results presentation were quite vague in presentation i.e. the tables and descriptions. Please refer to more specific comments below:

- Awareness was mentioned in some parts of the results and conclusion. However, there were no questions on awareness in the questionnaire. Please clarify.

- An example of vague sentence - 'Education shows a significant difference in the knowledge and attitude dimensions;. Please revise.

- Response rate was not reported. As this is an online survey, I would expect to see a response rate. This is where it is important that target populations are identified at the early stage of the study.

- 40.9% of the respondents were students. This could be due to chance but this is why it is important that ttarget populations are identified.

5. Conclusion - some sentences were ambiguous. Please rephrase.

Thank you and all the best.

6. PLOS authors have the option to publish the peer review history of their article (what does this mean?). If published, this will include your full peer review and any attached files.

Reviewer #1: No

Reviewer #2: No

Reviewer #3: No

---

## [Author Response · Author response to Decision Letter 0]

15 Feb 2022

Point-by-point reply and rebuttal to the reviewer’s comments 

Manuscript ID: PONE-D-21-12498

COVID-19 related knowledge, attitudes, and practices in Indian Population: An online national cross-sectional survey

Editor Comment: 

Comment-1 Please ensure that your manuscript meets PLOS ONE's style requirements, including those for file naming. The PLOS ONE style templates can be found at 

Reply: We are thankful to the reviewer in finding our manuscipt relevant for giving new insights and find it suitable for publication. Manuscript content is cross checked for PlOS One style amenability.

Comment-2. We suggest you thoroughly copyedit your manuscript for language usage, spelling, and grammar. If you do not know anyone who can help you do this, you may wish to consider employing a professional scientific editing service. 

Reply: The manuscript content is checked for grammar and spelling by the two independent researchers. 

The name of the colleague or the details of the professional service that edited your manuscript.

Prof.V.R.Rao MSc; PhD

Presently, Director (Research), Genome Foundation, Hyderabad;India

Visiting Professor, Thalassemia & Sickle Cell Society, Hyderabad;India

Former,ICMR Emeritus Medical Scientist

Dept of Genetics, Osmania University, Hyderabad;India

Former Professor, Biochemical Genetics & Molecular Anthropology Laboratory,

Department of Anthropology, Delhi University, Delhi; India

Email: profraovr@gmail.com

Comment-3. Please include additional information regarding the survey or questionnaire used in the study specifically please include a copy, in the original language

Reply: The details of questionnaire used in the Study is provided in Methods section and copy of questionnaire is its original language (English) is submitted under section: supporting Information.

Comment-4. PLOS requires an ORCID iD for the corresponding author in Editorial Manager on papers submitted after December 6th, 2016. Please ensure that you have an ORCID iD and that it is validated in Editorial Manager. To do this, go to ‘Update my Information’ (in the upper left-hand corner of the main menu), and click on the Fetch/Validate link next to the ORCID field. This will take you to the ORCID site and allow you to create a new iD or authenticate a pre-existing iD in Editorial Manager. Please see the following video for instructions on linking an ORCID iD to your Editorial Manager account: https://www.youtube.com/watch?v=_xcclfuvtxQ

Reply: The suggestion has been incorporated. 

Comment-5. Please ensure that you refer to Figure 3 in your text as, if accepted, production will need this reference to link the reader to the figure.

Reply: The suggestion has been incorporated in the manuscript. 

Comment-6. We note that Figure 2 in your submission contain map images which may be copyrighted. All PLOS content is published under the Creative Commons Attribution License (CC BY 4.0), which means that the manuscript, images, and Supporting Information files will be freely available online, and any third party is permitted to access, download, copy, distribute, and use these materials in any way, even commercially, with proper attribution. For these reasons, we cannot publish previously copyrighted maps or satellite images created using proprietary data, such as Google software (Google Maps, Street View, and Earth). For more information, see our copyright guidelines: http://journals.plos.org/plosone/s/licenses-and-copyright.

Reply: We agree with the editor comments and a new Map developed with freely available (QGIS) software has replaced the previous map. 

Comment-7. We note you have included a table to which you do not refer in the text of your manuscript. Please ensure that you refer to Table 4 in your text; if accepted, production will need this reference to link the reader to the Table.

Reply: The suggestion has been incorporated. 

Comment-8. Please include captions for your Supporting Information files at the end of your manuscript, and update any in-text citations to match accordingly. Please see our Supporting Information guidelines for more information: http://journals.plos.org/plosone/s/supporting-information. 

Reply: The suggestion has been incorporated. 

Reviewers' comments:

Reviewer #1: 

Comment-1: I did not see any reliability and validity test. We should know the Cronbach alpha value for this questionnaire. 

Reply: In the study, due to the mix (quantitative and qualitative) nature of KAP (Knowledge, attitude and practice) questionnaire the reliability and validity analysis could not be performed. In the questionnaire, questions with more than one valid answer and open-ended questions were also used, so we have not carried out the questionnaire internal consistency analysis using the statistics tools. However, questionnaires’ questions clarity and validity were tested in a pilot population (N=10). The question and domain were also critically reviewed by the social scientist, epidemiologist, and medical doctor for its suitability, applicability, relevance and accuracy. 

Comment-2 Would you please Provide the flow chart of your research? 

Reply: The flow chart could not be provided to minimize the manuscript length. Please see the below image: 

Comment-3 I did not see the objective in the abstract, And in methodology, please mention the exclusion criteria.

Reply: The suggestion has been incorporated in the manuscript. 

Comment-4 In the result, please mention in of table below the statistic that you used. I saw some paragraphs did not mention the table

Reply: The suggestion has been incorporated in the manuscript.

Comment-5 For discussion, please add more the similar study in others country.

Reply: suggestion has been incorporated and recent studies has been updated in manuscript.

Comment-6 In conclusion, please make sure the statement based on the objective of the study

Reply: The suggestion has been incorporated in the manuscript. Additional details has been deleted from conclusion.

Reviewer #2: 

Overall found this to be a good read and well-done research taking into account the circumstances of the pandemic at the time of the study. That shows forward-thinking and adaptability.

Reply: We are thankful to the reviewer in finding our ms relevant for giving new insights and find it suitable for publication.

Comment-1 Introduction

second paragraph. Kindly add the year 2020 to the 27th January to prevent any ambiguity regarding the time although it is apparently obvious. It does not hurt to add it.

Last sentence of paragraph 2; would be good to have the figures as of the time of this article being published or a recent update in 2021. Also "....as on August..." should be "...as of August.."

Reply: The suggestion has been incorporated in the manuscript

Comment-2 third paragraph.

Kindly rephrase this to reflect that you refer to that period. Maybe you could change "so far" to "At that time" and "have been developed" to "had been developed" as well as the subsequent tense of the sentence to match.

Reply: The suggestion has been incorporated in the manuscript. The part has been deleted, as its not reflect the real situation. 

Comment-3 Regarding the situation during the study period, kindly change the tense to the past tense like above to reflect what was happening at that time. In this current state it sounds like that is the current situation in India.

Reply: The suggestion has been incorporated in the manuscript. The part has been deleted, as its not reflect the real situation. 

Comment-4 METHODS

Line 6: participants were also asked to share the survey link with "their" instead of "his"

Reply: The suggestion has been incorporated in the manuscript.

Comment-5 Questionnaire

line 5: you mention infection status. I think this would be better if defined as it leaves room for subjectivity. What does this mean for a participant who selects yes or no. Is their answer based on official results from the country's testing program or from symptoms the patient has or some other means.

Reply: We agree with the reviewer comments and the requited changes has been made in the manuscript. During the study period, the self-volunteer as well as state regulated testing of suspected individuals has been done by state authority. 

Comment-6 Scoring and analysis

State the basis of the scoring system used. Was it arbitrarily generated or based on for example WHO's questions you developed your questionnaires on? It is important that this is very clear as your entire study seems to be based on this scoring system. the scores are mentioned again in a manner i believe not to be intuitive. Clarifying the scoring system will help this. Also state what the overall score would have been again to make it easy for the reader to follow. 

Reply: The scoring system followed in the study was inspired by the recently published study with a similar design cited in the manuscript (Orimbo, 2020). WHO guidelines were followed for the theme identification and structure of the questionnaire. For the scoring system, one point was awarded for each rightly answered question (for questions with only one correct option) and option (for questions which had more than one correct option) and no point was awarded for wrongly answered questions/options. We defined the criteria for the assessment of knowledge, attitude, and practices. A cutoff score of >=median value within each dimension was used to categorize the outcome as low, and high (for knowledge and awareness), positive and negative (for attitudes) and good and poor (for practice). The analysis was followed by the recently published study. Some addition has been made in the section for easy understanding of readers. 

Orimbo EO, Oyugi E, Dulacha D, Obonyo M, Hussein A, Githuku J, Owiny M, Gura Z. Knowledge, attitude and practices on cholera in an arid county, Kenya, 2018: A mixed-methods approach. PloS one. 2020 Feb 26;15(2):e0229437.

Comment-7 RESULTS, Kindly review this in the results section.

KNowledge and Awareness assessment

Line 7: the results demonstrate that knowledge scores.... The first part can be removed to make the sentence less wordy and read better. It could be "knowledge scores were found to be high in the age group of 31 and above, female gender, working participants, individuals with higher education, and people living in the urban area."

Reply: Suggestion has been incorporated in the manuscript. 

Comment-10 Discussion

The first paragraph provides context and the need for your work. This belongs in the introduction as justification. The discussion should begin with key findings of the study

Reply: Suggestion has been incorporated in the manuscript and additional details has been deleted from discussion part. 

Comment-11 Limitations should also come before the conclusions

Reply: Suggestion has been incorporated. 

Reviewer #3: Dear Authors,

Please kindly refer to my specific comments for improvement of the manuscript below:

Comment-1 1. Keywords in abstracts - There are a few keywords that I find unnecessary e.g. awareness and survey. Please kindly revise

Reply: Suggestion has been incorporated. 

Comment 2. Spelling - please standardize the spelling for 'coronavirus' vs 'corona virus'.

Reply: Suggestion has been incorporated in the manuscript. 

Comment-3. Methodology - Please refer to the questions below:

- Was there any target population planned before the distribution of the survey? I understand that the survey was distributed in Facebook and Whatsapp. However, there must be target groups that you were trying to reach. Please clarify.

Reply: In the study, the population targeted includes the characteristics of adult (Above 18 year, living in Indian territory and understand the survey language (Literate). 

Comment-3.1- Has the questionnaire undergone reliability and validity tests? If yes, please include the Cronbach's alpha values.

Reply: In the study, due to the mix (quantitative and qualitative) nature of KAP (Knowledge, attitude and practice) questionnaire the reliability and validity analysis could not be performed. In the questionnaire, questions with more than one valid answer and open-ended questions were also used, so we have not carried out the questionnaire internal consistency analysis using the statistics tools. However, questionnaires’ questions clarity and validity were tested in a pilot population (N=10). The question and domain were also critically reviewed by the social scientist, epidemiologist, and medical doctor for its suitability, applicability, relevance and accuracy. 

Comment-3.2- Who performed the questionnaire translation? Did the translation process follow a standard protocol? Please elaborate.

Reply: The translation of questionnaire was performed by expert team of Rajbhasha (Hindi) division, National institute of Malaria research. The Hindi translated version were retranslated to original (English) version, following the backward translation. The two versions were compared for content similarity and no significant difference were observed. The details has been incorporated in the manuscript. 

Comment-3.3- Please revise the total number of questions to 45 and not 56 as the other 11 questions were related to demographics.

Reply: Suggestion has been incorporated in the manuscript.

Comment-3.4- What does the term 'dropout' response refer to here? I personally think that the term is not applicable in survey-type research.

Reply: Suggestion has been incorporated and term dropout has been omitted from the manuscript. 

Comment-3.5- What was the formula used to calculate the sample size in this research? Please clarify and think of the need to calculate the sample size for this type of research.

Reply:

The study followed the suggested survey design research formula to calculate the sample size. (Charan, 2013). The formal sample size calculation was performed considering the adult population number of India, and the response rate on the KAP study. The proportion of the population having adequate knowledge regarding COVID-19 was assumed to be 50% as no similar study was conducted in the study population, level of significance 5%, and margin of error 5% were considered for calculation and target sample size for the survey was calculated as 385 using formula (“Sample Size = (Distribution of 50%) / ((Margin of Error%/ Confidence Level Score)2)”).

Charan J, Biswas T. How to calculate sample size for different study designs in medical research?. Indian journal of psychological medicine. 2013 Apr;35(2):121-6.

Comment-4. Results - the overall results presentation were quite vague in presentation i.e. the tables and descriptions. Please refer to more specific comments below:

Reply: Suggestion has been incorporated in the manuscript.

Comment-4.1- Awareness was mentioned in some parts of the results and conclusion. However, there were no questions on awareness in the questionnaire. Please clarify.

Reply: The first dimension of the questionnaire was Knowledge and awareness, so accordingly the term was used in the manuscript. 

Comment-4.2- An example of vague sentence - 'Education shows a significant difference in the knowledge and attitude dimensions; Please revise.

Reply: Suggestion has been incorporated in the manuscript.

Comment-4.3- Response rate was not reported. As this is an online survey, I would expect to see a response rate. This is where it is important that target populations are identified at the early stage of the study.

Reply: A total of 527 responses were received out of which 13 participants denied consent, ten responses were duplicate, and three were from non-Indians. Following the study protocol, a total of 501 qualified the study participants’ criteria and completed the questionnaire and hence were included in the analysis. Considering the mentioned numbers, proportion of 95.06% response can be calculated. The actual response rate could not be reported, as the survey link was shared from different end to many individuals which cannot be traced. 

Comment-4.4- 40.9% of the respondents were students. This could be due to chance but this is why it is important that ttarget populations are identified.

Reply: The high proportion of higher educated students in the study population reported due to the convenient nature of sampling, internet accessibility and familiarity of the survey mode of respondents. 

Comment-5. Conclusion - some sentences were ambiguous. Please rephrase.

Reply: Suggestion has been incorporated in the manuscript.

_________

---

## [Editor Report · Decision Letter 1]

17 Feb 2022

COVID-19 related knowledge, attitudes, and practices in Indian Population: An online national cross-sectional survey

PONE-D-21-12498R1

Dear Dr. Piyoosh Kumar Singh,

We’re pleased to inform you that your manuscript has been judged scientifically suitable for publication and will be formally accepted for publication once it meets all outstanding technical requirements.

Kind regards,

Muhammad Junaid Farrukh

Academic Editor

PLOS ONE
---

## [Editor Report · Acceptance letter]

21 Feb 2022

PONE-D-21-12498R1 

COVID-19 related knowledge, attitudes, and practices in Indian Population: An online national cross-sectional survey 

Dear Dr. Singh:

I'm pleased to inform you that your manuscript has been deemed suitable for publication in PLOS ONE. Congratulations! Your manuscript is now with our production department. 

Kind regards, 

on behalf of

Dr. Muhammad Junaid Farrukh 

Academic Editor

PLOS ONE